# EXTRACTING RULE-BASED DESCRIPTIONS OF ATTENTION FEATURES IN TRANSFORMERS

## ABSTRACT

Mechanistic interpretability strives to explain model behavior in terms of bottom-up primitives. The leading paradigm is to express hidden states as a sparse linear combination of basis vectors, called features. However, this only identifies which text sequences (exemplars) activate which features; the actual interpretation of features usually requires subjective and time-consuming inspection of these exemplars. This paper advocates for a different solution: *rule-based descriptions* that match token patterns in the input and correspondingly increase or decrease the likelihood of specific output tokens. Specifically, we extract rule-based descriptions of SAE features trained on the outputs of *attention* layers. While prior work treats the attention layers as an opaque box, we describe how it may naturally be expressed in terms of interactions between input and output features, of which we study three types: (1) skip-gram rules of the form "*[Canadian city] ... speaks → English*", (2) absence rules of the form "*[Montreal] ... speaks ↛ English*," and (3) counting rules that toggle only when the count of a word exceeds a certain value or the count of another word. Absence and counting rules are not readily discovered by inspection of exemplars, where manual and automatic descriptions often identify misleading or incomplete explanations. We then describe a simple approach to extract these types of rules automatically from a transformer, and apply it to GPT-2 small. We find that a majority of features may be described well with around 100 skip-gram rules, though absence rules (2) are abundant even as early as the first layer (in over a fourth of features). We also isolate a few examples of counting rules (3). This paper lays the groundwork for future research into rule-based descriptions of features by defining them, showing how they may be extracted, and providing a preliminary taxonomy of some of the behaviors they represent.

## 1 INTRODUCTION

A growing body of work under the umbrella of *mechanistic interpretability* seeks to dissect and understand the behavior of transformer language models (LMs). Starting from the investigation of specific computational motifs like induction heads (Elhage et al., 2021; Olsson et al., 2022), research in this area has largely converged to two popular methods of studying a transformer. First, by isolation of a "circuit" formed by MLP layers and attention heads that implements a specific trait (e.g., addition) (Meng et al., 2022; Geva et al., 2023)—and second, by the use of Sparse Autoencoders (SAEs) to extract feature vectors from the transformer hidden states (Bricken et al., 2023; Huben et al., 2024). The latter has become the predominant paradigm, but the interpreta-

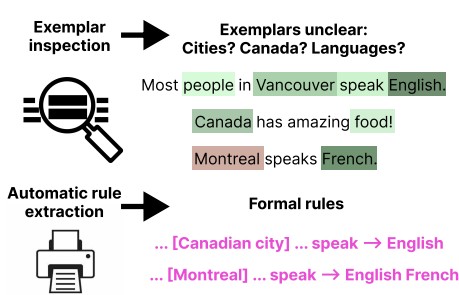

Figure 1: SAE features are usually explained via examplars, which are subjective and hard to interpret. We approximate them with *rules* that promote or suppress the feature.

tion of feature vectors still relies on manual (or LM-assisted) inspection of exemplars with high feature activations (Fig. 1). Such interpretation is subjective and often incomplete or inaccurate. Polysemanticity (Elhage et al., 2022) of features further frustrates attempts to interpret feature exemplars. There have also been some preliminary attempts to extract circuits of SAE features: graphs

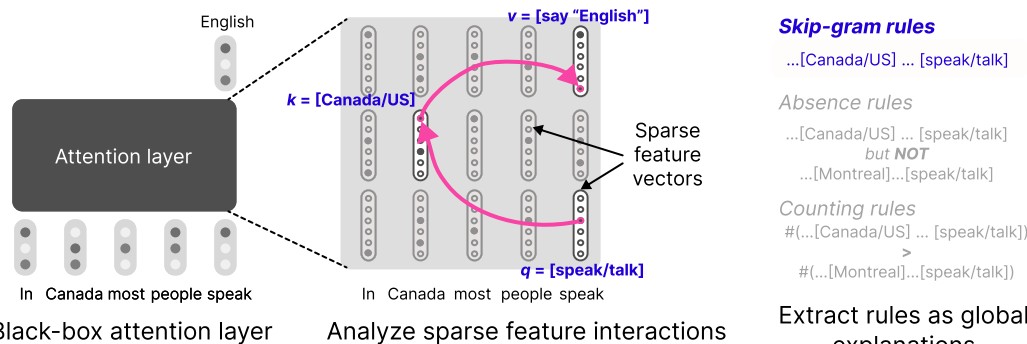

Figure 2: Given an attention layer in a transformer language model (*left*), our goal is to explain each output feature as an explicit function of input features (*center*), and provide a global description of these functions in terms of formal rules (*right*).

specifying how features from lower layers combine to activate features at higher layers (Marks et al., 2024; Dunefsky et al., 2024b; Ge et al., 2024; Ameisen et al., 2025). However, these methods have treated the attention component of the transformer as a constant, explaining how features interact in a given prompt based on the observed attention pattern, but they do not explain how the attention pattern itself arises from lower-layer features.

In this paper, we advocate for a different approach—we study the same SAE features as prior work, but extract inherently interpretable descriptions that take the form of *symbolic, human-readable rules*. Our task is to prescribe a framework of rules that is expressive enough to approximate the computation in a transformer, but also enables efficient extraction of said rules from it. We tackle this challenge in the context of the attention layers in a transformer. Assuming that one has already extracted features from the hidden states before and after the attention layer, we express the head's computation as a weighted sum over attention between features from one position to another (Figure 2). Each term can be interpreted as matching a template $[a] \ldots [b]$ in the input, and the weights correspond to increasing or decreasing the likelihood of the corresponding output feature. We identify three types of rules from these interactions: (1) *skip-gram rules* of the form "*[Canadian city] ... speaks → English*" that promote "*English* when the pattern *[Canadian city] ... speaks*" is observed, (2) *absence rules* "*[Montreal] ... speaks ↛ English*" that suppress the production of *English* following the pattern, and (3) *count-based rules* that arise from competition between rules of the form (1) and (2), and produce a token only when a certain pattern is more frequent than another. It is noteworthy that absence and counting rules are not easily discoverable by inspecting exemplars alone.

We present an empirical pipeline to extract these features from SAEs trained on an attention head automatically. Upon deployment of this pipeline to GPT-2 small (Radford et al., 2019), we find that skip-gram features already achieve a good approximation of several attention heads—especially in early layers. We also identify cases where they fail: later layers require longer rules, and we find sophisticated behaviors such as distractor suppression (where the presence of one feature inhibits the expected response to another) and counting operations (where outputs depend on the frequency rather than mere presence of input features). In fact, both behaviors are observed as early as the first layer—with over a third of skip-gram features being accompanied by at least one distractor. Our experiments demonstrate clear potential for rule-based features to explain language model behavior. We hope future research builds on the groundwork laid here to extract a more complete set of rules.

## 2 BACKGROUND

**Transformer language models.** The transformer (Vaswani et al., 2017) is a neural network architecture for processing sequences. It consists of a composition of feed-forward layers, which process features at a single position in the sequence, and multi-headed attention layers, which combine information from multiple positions in the sequence. We focus on decoder-only transformer language models (LMs), which take a sequence of tokens as input and output a probability distribution over the next token. Our focus in this work is on individual heads in the attention layer, which map a sequence

of embeddings $\mathbf{x}_1, \ldots, \mathbf{x}_t \in \mathbb{R}^{d_{\text{model}}}$ to a sequence $\mathbf{y}_1, \ldots, \mathbf{y}_t \in \mathbb{R}^{d_{\text{head}}}$, where the output is given by:

$$\mathbf{y}_t = \sum_{i \leq t} a_{t,i} \mathbf{W}_V \mathbf{x}_i \qquad a_{t,i} = \frac{\exp(\mathbf{x}_t^\top \mathbf{W}_Q^\top \mathbf{W}_K \mathbf{x}_i)}{\sum_{i \leq t} \exp(\mathbf{x}_t^\top \mathbf{W}_Q^\top \mathbf{W}_K \mathbf{x}_i)}$$

The weights $\mathbf{W}_K, \mathbf{W}_Q, \mathbf{W}_V \in \mathbb{R}^{d_{\text{head}} \times d_{\text{model}}}$ are referred to as the *key*, *query*, and *value* projection matrices, respectively; $a_{t,i}$ denotes the *attention* from position $t$ to position $i$; and $d_{\text{head}} = d_{\text{model}}/h$, where $h$ is the number of attention heads per layer.

**Sparse autoencoder.** Each input $\mathbf{x}_i$ can be decomposed into a weighted sum over *input features*[1]:

$$\mathbf{x}_i = \sum_{j=1}^{n} f_j(\mathbf{x}_i) \mathbf{d}_j + \bar{\mathbf{x}}_i, \tag{1}$$

where the *input activations* $f_j(\mathbf{x}_i) \triangleq \sigma(\mathbf{x}_i^T \tilde{\mathbf{d}}_j) \in (0,1)$ are given by the sigmoid of its projection along an up-projection vector associated with feature $j$. We will omit the irreducible error $\bar{\mathbf{x}}_i$ in subsequent discussions since our analysis does not depend on it. The vectors $\mathbf{d}_j, \tilde{\mathbf{d}}_j$ are jointly trained to minimize a reconstruction loss between the two sides of Equation 1 subject to a sparsity penalty. Interested readers may refer to, e.g., Bricken et al. (2023) for more details. Similarly, we can assume that $\mathbf{y}_t$ has been decomposed into a weighted sum over *output features* $\{\mathbf{u}_j, \tilde{\mathbf{u}}_j\}_{j=1}^{n}$ (with corresponding *output activations*).

## 3 RULE-BASED DESCRIPTIONS OF ATTENTION FEATURES

In this section, we rewrite the computation of an attention head as a weighted sum of terms that represent the promotion or suppression of output tokens based on interactions of feature pairs. We then advance an interpretation of this interaction

### 3.1 DECOMPOSING ATTENTION FEATURES

**Expressing output activations in terms of input activations.** When is $\sigma(\mathbf{y}_t^T \mathbf{u})$ high for an output feature $\mathbf{u}$? We start by rewriting the output activation in terms of the input activations:

$$\sigma(\mathbf{y}_t^T \mathbf{u}) = \sigma\left( \sum_{i \leq t} a_{t,i} \mathbf{x}_i^T \mathbf{W}_V^T \mathbf{u} \right) = \sigma\left( \sum_{\substack{i \leq t \\ j=1}}^{n} a_{t,i} f_j(\mathbf{x}_i) \mathbf{d}_j^T \mathbf{W}_V^T \mathbf{u} \right) \equiv \sigma\left( \sum_{\substack{i \leq t \\ j=1}}^{n} a_{t,i} f_j(\mathbf{x}_i) \mathbf{d}_j^T \mathbf{W}_V^T \mathbf{u} \right)$$
$$\tag{2}$$

where the attention weights $a_{t,i}$ can themselves be expressed in terms of the input activations:

$$a_{t,i} \propto \exp\left( \mathbf{x}_t^\top \mathbf{W}_Q^\top \mathbf{W}_K \mathbf{x}_i \right) = \exp\left( \sum_{j,k=1}^{n} f_j(\mathbf{x}_t) f_k(\mathbf{x}_i) \mathbf{d}_j^\top \mathbf{W}_Q^\top \mathbf{W}_K \mathbf{d}_k \right) \tag{3}$$

For ease of exposition, we introduce the following shorthand notation:

$$\mathcal{S}(j) = \mathbf{d}_j^\top \mathbf{W}_V^\top \mathbf{u}, \;\; \mathcal{A}(j,k) = \mathbf{d}_j^\top \mathbf{W}_Q^\top \mathbf{W}_K \mathbf{d}_k.$$

**Interpretation of Equations 2 and 3.** Equation 2 tells us that each pair $(\mathbf{d}_j, \mathbf{u})$ of input-output features either promotes or discourages the output to lie along $\mathbf{y}_t$, depending on whether the sign of $\mathcal{S}(j)$ is positive or negative. These two cases may be interpreted as rules $[\mathbf{d}_j] \ldots [\mathbf{u}] \rightarrow \mathbf{y}_t$ or $[\mathbf{d}_j] \ldots [\mathbf{u}] \nrightarrow \mathbf{y}_t$. These rules depend recursively upon the descriptions of $\mathbf{d}_j$ and $\mathbf{u}$, and are abstract versions of **skip-gram** and **absence** rules. We illustrate these rule types in Figure 3. We call these output-value rules since they arise from the output-value interaction in Equation 2. Further, the extent of the promotion or suppression depends on several factors.

---

[1]Please note the terminology: we always use input/output *embeddings* to refer to $\mathbf{x}_i, \mathbf{y}_t$, and *features* to refer to $\mathbf{d}_j, \mathbf{u}_j$. *Activations* refer to the sigmoid-rectified projections onto $\tilde{\mathbf{d}}_j, \tilde{\mathbf{u}}_j$.

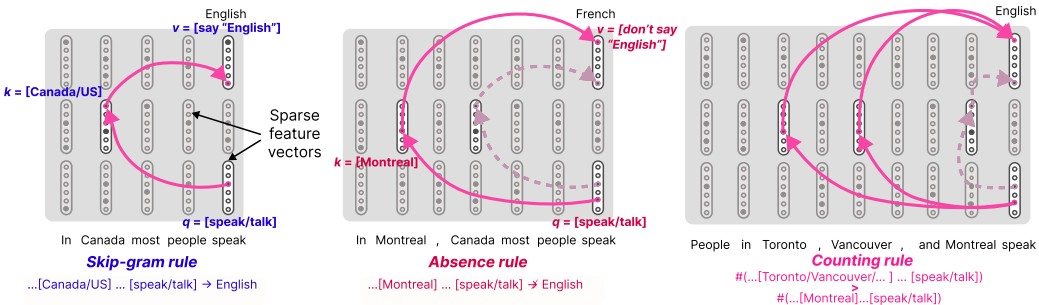

Figure 3: Attention rules can take different forms, depending on how the input features interact. Query-key interaction may promote the generation of specific outputs, leading to *skip-gram* rules (*left*). Analogously, suppression leads to *absence* rules (*center*). Competition between the two types may lead to *counting rules* (*right*).

1. It depends upon the input activation $f_j(\mathbf{x}_i)$ of tokens $\mathbf{x}_i$ along feature $j$. Features more strongly exhibited by $\mathbf{x}_i$ affect the output more.

2. It is modulated by the strength of the attention $a_{t,i}$ from position $t \to i$. This strength in turn depends on *query-key* interactions between features $\mathbf{d}_j$ and $\mathbf{d}_k$. Note that query-key interaction can modulate the attention score both upward (promotion) and downward (suppression), and depends on the input activations $f_j(\mathbf{x}_i)$ once again. These correspond to the skip-gram rule $[\mathbf{d}_k] \ldots [\mathbf{d}_j] \to \mathbf{y}_t$ and absence rule $[\mathbf{d}_k] \ldots [\mathbf{d}_j] \not\to \mathbf{y}_t$, respectively.

3. The sum in Equations 3 implies that two different interactions may simultaneously try to promote and suppress the output $\mathbf{y}_t$. This leads to **counting rules** where the net effect depends upon the relative counts of two distinct interactions.

We note that Ge et al. (2024) perform a similar analysis of attention in terms of SAE features. Interestingly, skip-gram and absence rules may surface both through the output-value interaction in Equation 2 and query-key interaction in Equation 3. Skip-gram rules were also considered by Elhage et al. (2021).

## 3.2 EXTRACTING ATTENTION RULES

This subsection makes the notion of skip-gram and absence rules more concrete, and describes a simple heuristic to find the most salient query-key rules for a given attention head.

**Mapping input features to token patterns.** Consider the case where the head lies in the first layer of the transformer. As each input embedding corresponds to a unique token, query-key rules can be mapped directly to token-level skip-gram absence rules, e.g., "*Vancouver ... speaks → English*." On the other hand, output-value rules depend upon the output feature $[\mathbf{u}]$, for which we do not possess a formal description. We do not ascribe tokens to these features in this paper, but paths forward include falling back upon exemplar-based descriptions, or iterative refinement of the description based on the currently identified key-key and query-key features. This approach can also describe attention features in later layers of an attention-only transformer in terms of skip-gram and absence-rules of the previous layer. It leads to a growing list of rules with increasing complexity.[2] In transformers with MLP layers, one can obtain approximate (but incomplete) descriptions by ignoring them.

**Ranking query-key rules.** Now that we have formal descriptions of the query-key interactions, we wish to find the interactions that maximally contribute to Equation 3. The number of key-query interactions is quadratic in the size of the input SAE dictionary, and we would like to bring this number down. We propose two heuristics motivated from magnitude-based and gradient-based pruning methods, (e.g. Voita et al., 2019; Michel et al., 2019). We first pick the 100 key features $j$ with the highest values of $\mathcal{S}(j)$. Then, for each of these key features $j$, we pick the 100 query features maximizing $\mathcal{A}(j, k)$, resulting in 10,000 key/query pairs.

---

[2]For example, $[a] \ldots [b] \ldots [c] \to d$.

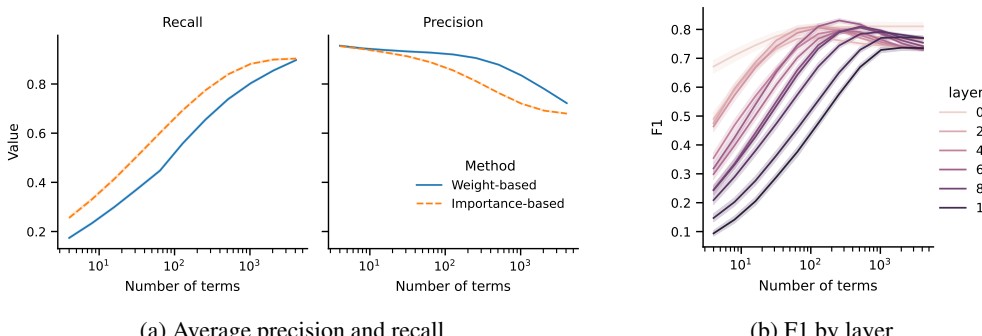

(a) Average precision and recall          (b) F1 by layer

Figure 4: Precision and recall for predicting binarized activation values using skip-gram rules, as a function of the maximum number of terms (meaning pairs of key and query features) per output feature. In Fig. 4a, terms are selected either according to the magnitude of their weights (*Weight-based*) or by calculating a gradient-based importance score using a small training set (*Importance-based*; see Sec. 4 for more details). Results are averaged over 100 features for each head in GPT-2 small. Importance-based scoring achieves higher recall with fewer terms, but with lower precision. Fig. 4b shows the F1 scores by layer, selecting terms according to importance. Higher layers generally have worse approximation scores and require more terms.

- Weight-based: we simply sort the list of pairs in decreasing order of $\mathcal{A}(j, k) \times \mathcal{S}(j)$.
- Gradient-based: we introduce an auxiliary variable $m_{j,k} = 1$ for each $(j, k)$ pair, and redefine $\mathcal{A}(j, k) = m_{j,k} \mathbf{d}_j^\top \mathbf{W}_Q^\top \mathbf{W}_K \mathbf{d}_k$ for use in Equation 3.

We then sort each rule $(j, k)$ in descending order of $\frac{\partial \sigma(\mathbf{y}_t^T \mathbf{u})}{\partial m_{j,k}}$ which represents the importance of the edge $k \to j$ for the output feature activation. For a given output feature, there might be multiple features with positive value scores, and multiple queries that attend to those keys. In that case, we consider the description of the output feature to be the disjunction $\bigvee_i^N r_i$ of the individual rules $r_i$.

## 4 EXPERIMENTS

In the previous section, we discussed different types of rules that attention heads can express. In this section, we evaluate how well these rules approximate attention features in practice.

**Setup.** We train SAEs on the output of every attention head in GPT-2 small (to extract the output features of Section 3.2). Our training setup follows Kissane et al. (2024), with the exception that we train SAEs on each attention head individually (rather than training one SAE on the concatenated outputs). Specifically, we train the SAEs on sequences of 64 tokens from OpenWebText (Gokaslan et al., 2019), following Kissane et al. (2024). We use pretrained SAEs from Bloom (2024) to extract input features. For our evaluation, we collect feature activations for 50,000 sequences from the training data, following Bills et al. (2023). For each attention head, we randomly sample 100 features that are active in at least 100 sequences. Similar to Foote et al. (2023), we evaluate the rules at the binary task of predicting whether or not the feature is active for a given prefix, and report precision, recall, and F1 scores. If the rule does not predict any positive examples, we assign a precision of 1. For each feature, we use the 100 prefixes with the highest activations as positive examples and randomly sample 100 prefixes with an activation of 0 to serve as negative examples, randomly split into equal-sized training and evaluation sets. We provide additional details in Appendix A.1.

### 4.1 SKIP-GRAM RULES

We start by evaluating skip-gram rules, as described in Sec 3.2, comparing weight-based and gradient-based methods for identifying a small number of key/query pairs for a given output feature. Given a set of key/query pairs for a particular output feature, we predict that the output feature will be active for a token $\mathbf{x}_t$ if there is a query feature $q$ with $f_q(\mathbf{x}_t) > 0$ and a key feature $k$ with $f_k(\mathbf{x}_{t'}) > 0$ for any $t' \leq t$ such that attention-score$(q, k) > 0$ and value-score$(k) > 0$.

| DFA | Feature activations |
|---|---|
| as soon as possible.\n\nIf that's his attitude, then | as soon as possible.\n\nIf that's his attitude, then |
| cremated.\n\nIf a funeral home is not moving the body , then | cremated.\n\nIf a funeral home is not moving the body , then |
| your game data first.\n\nIf you want to take the risk, then | your game data first.\n\nIf you want to take the risk, then |

(a) Top activating sequences for L0H0.94.

| Key | Val. score | Query | Attn. score |
|---|---|---|---|
| If | 0.139 | _then | 0.132 |
| | | _Then | 0.089 |
| | | _you | 0.088 |
| Both | 0.108 | Ã©n | 0.076 |
| | | pos69 | 0.074 |
| | | pos81 | 0.073 |

(b) Top scoring key/query pairs.

Figure 5: Sequences that activate a layer-0 attention feature in GPT-2 small (5a) and the highest-scoring pairs of key and query input features associated with this feature (5b). The three examples can be explained by the presence of a single skip-gram pattern, `...[If]...[then]`; see Section 4.1.

**Results.** Fig. 4 plots the precision, recall, and F1 for the predicted activations for all features, aggregating the features from all layers and heads. Fig. 4a shows that skip-grams provide a relatively good approximation, and improve with additional terms. The gradient-based method for selecting input features achieves higher recall with fewer terms. This suggests that rule-extraction could be a feasible approach to explaining transformer features, even using a simplified form of rule. Fig. 4b shows that higher-layer features tend to have lower approximation scores and require more terms, which could be due to limitations of the underlying feature decomposition.

**Qualitative analysis.** Fig. 5 shows an example of an attention output feature from one of the attention heads in the first layer of GPT-2 small. In Fig. 5a, we show three prefixes that have high activations for this feature. One column shows the feature activations for each token, and the other shows the direct feature attribution (DFA), following Kissane et al. (2024). The DFA score for a token $\mathbf{x}_{t'}$ is the influence of that token on the output feature activation: $\text{DFA}(\mathbf{x}_t) = a_{t,t'} \mathbf{x}_{t'}^\top \mathbf{W}_V^\top \mathbf{u}$, where $\mathbf{u}$ is the SAE encoder vector associated with the output feature and $a_{t,t'}$ is the attention assigned to position $t'$ from position $t$. Fig. 5b shows the highest-scoring pairs of key and query features. Because the input features are taken at the first layer, we can interpret them by identifying the input tokens that have the highest feature activations. Consistent with the examples, the highest scoring feature pair consists of key tokens like "If" and query tokens like "then," which we can represent as the skip-gram pattern `...[If]...[then]`, and expect that the feature activates when this pattern is present in the sequence. In this example, a single key/query pair explains the majority of the observed behavior. We show additional examples in Appendix A.4, including a feature that activates in more diverse contexts (App. Fig. 8b). In this case, we can define the rule as a disjunction of skip-grams, expecting the feature to fire if the sequence contains any of the skip-grams in the list.

## 4.2 Absence Rules

Next, we investigate whether our attention features represent "absence rules," as described in Sec. 3.2. To find whether a feature encodes an absence rule, for each output feature $g$, we identify the input feature $k$ with the highest value score and the input feature $q$ with the highest attention score with $k$. Then we check if there is another input feature $k'$ with attention-score$(q, k') >$ attention-score$(q, k)$ and value-score$(k', g) < 0$. As a rough estimate, we consider only the highest-scoring $k, q$ pair for each feature, and check whether there is a distractor key $k'$.

**Results.** We find that, for the majority of skip-gram features, there is a distractor feature $k'$ satisfying the description above (see Appendix Sec. A.4 for the statistics). Fig. 6a shows that a small but significant number of examples that contain a skip-gram pattern `...[k]...[q]` also contain the distractor key, and Fig. 6b shows that examples containing the distractor key tend to have lower activations than examples containing only the skip-gram features. To further validate that the distractor feature $k'$ suppresses the activation of $g$, we create a counter-factual dataset as follows: given a distractor $k'$, we find the token $w'$ that maximally activates $k'$: $w' = \text{argmax}_{w' \in \mathcal{V}} f_{k'}(\mathbf{e}_{w'})$, where $f_{k'}$ is the feature activation for feature $k'$ in the input SAE and $\mathbf{e}_{w'}$ is the token embedding for

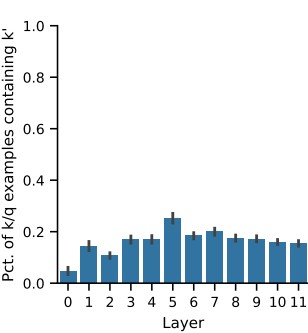 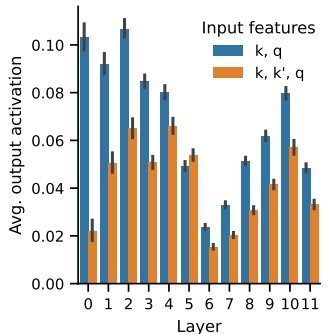 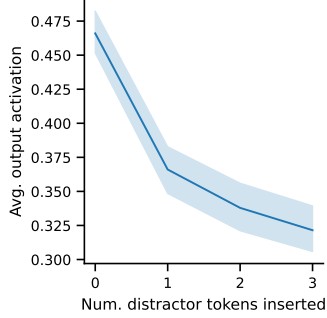

(a) Prevalence of inputs with distractor keys.

(b) Avg. activation with and without distractor keys.

(c) Avg. activation after adding distractor tokens.

Figure 6: Between 5 and 25% of examples that contain the skip-gram pattern `...[k]...[q]` also include a "distractor" feature $k'$, which attracts attention away from $k$ and suppresses the output activation (Fig. 6a).Inputs containing $k$, $q$, and $k'$ have lower output feature activations on average compared to inputs containing only $k$ and $q$ (Fig. 6b). To further validate the effect of $k'$, we find inputs that activate the feature and prepend a word that triggers the distractor key, finding that the output activation is lower as we add more copies of the distracting token (Fig. 6c).

token $w'$. (We limit this investigation to features in the first attention layer.) Then we find dataset examples that contain $k$ and $q$ and prepend $w'$ from one to four times. Fig. 6c plots the average result, showing that inserting the distractor feature does suppress the output feature. This suggests that, even though the distractor pattern might occur relatively rarely in naturalistic data, absence rules need to be taken into account in order to fully characterize the behavior of attention features.

**Qualitative analysis.** Appendix A.4 includes examples of the absence rules we recover. For example, Fig. 10 shows a feature with the skip-gram description `...[://]...[com]`, suggesting that the feature activates at the end of URLs. However, if the token "twitter" appears in the sequence, it attracts attention from the key token and suppresses the output feature, meaning that the activation of this output feature implicitly encodes the absence of the "twitter" feature. The resulting description resembles the "feature splitting" phenomenon described by Chanin et al. (2024), corresponding to URLs, except for URLs containing "twitter."

### 4.3 COUNTING RULES

The rules discussed above both assume that features can be explained by one-hot attention, where a query feature attends to a single key feature. However, some attention features might depend on attending broadly to multiple tokens in a sequence. In particular, attention heads can implement a "counting" feature that activates as a function of the number of times some input feature $k$ appears in the sequence (Fig. 3). For example, Weiss et al. (2021) and Liu et al. (2023) present constructions for how an attention head can count by using a beginning-of-sequence token as a kind of attention sink. We find that such rules do exist in GPT-2 small, even as early as the first layer (Figure 7). Attention heads can also use a broad-attention, counting construction to calculate more sophisticated arithmetic features by using interactions between value embeddings. For example, Yao et al. (2021) show how an attention head balances parentheses by calculating the difference between the number of open parentheses and the number of closed parentheses. Understanding attention features might therefore require understanding how value embeddings interact.

## 5 DISCUSSION

**Feature interactions.** The attention score between query token $\mathbf{x}_t$ and key token $\mathbf{x}_{t'}$ depends on the sum of a potentially quadratic number of feature interactions. In our analysis, we have assumed that rules can be defined in terms of a single feature at each token. For example, for the skip-gram rule `...[k]...[q]`, we assume that the feature will fire if feature $q$ is active at $\mathbf{x}_t$ and feature

*Max activation: 0.7943181991577148*

*DFA:*

<|endoftext|> up I often heard "What, you play video games?! That's so awesome, girls never play video games!", but now when you tell someone you play video games you'd sooner get the question what kind of games

*Feature activations:*

<|endoftext|> up I often heard "What, you play video games?! That's so awesome, girls never play video games!", but now when you tell someone you play video games you'd sooner get the question what kind of games

*Max activation: 0.6259156465530396*

*DFA:*

<|endoftext|> as buggy [as newer ones], but we♦♦re more tuned-in at looking for the bugs. I personally remember old PC games and even old Nintendo games that had tons of bugs. I think the big difference is that the core technology of games

*Feature activations:*

<|endoftext|> as buggy [as newer ones], but we♦♦re more tuned-in at looking for the bugs. I personally remember old PC games and even old Nintendo games that had tons of bugs. I think the big difference is that the core technology of games

*Max activation: 0.5767968893051147*

*DFA:*

<|endoftext|> best Wear OS games (Android Wear games

*Feature activations:*

<|endoftext|> best Wear OS games (Android Wear games

Figure 7: Sequences that activate an attention feature from head 10 in layer 0 in GPT-2 small. The activation tends to be higher when the word "games" appears multiple times in the sequence, indicating that this feature exhibits a counting rule.

$k$ is activate at $\mathbf{x}_{t'}$, regardless of the other features active at those tokens. However, the attention could depend on multiple key/query interactions. As illustration, one way this could arise is due to positional features. For example, we could imagine that if $\mathbf{x}_t$ and $\mathbf{x}_{t'}$ have one pair of features with a positive attention score (e.g. if $\mathbf{x}_{t'}$ = "If" and $\mathbf{x}_t$ = "then"), and another pair of features with a negative attention score, if $\mathbf{x}_{t'}$ occurs much earlier in the sequence than $\mathbf{x}_t$. Future work is needed to develop interpretable ways of describing such features, where the rule depends on the balance between the positive and negative feature interactions at each position.

**Underlying feature decomposition.** We have aimed to explain attention features in terms of features from a pre-trained SAE applied to the input layer. However, these features are not necessarily the best unit of anlaysis for characterizing attention rules. One direction for future work could be to explore SAE variants, such as transcoders (Dunefsky et al., 2024a) and other variants (e.g. Gao et al., 2025; Bussmann et al., 2024; Rajamanoharan et al., 2024), or to train SAEs on key and query embeddings. Perhaps a more promising approach is to develop methods for optimizing the input feature decomposition jointly with the attention output features, with regularization to favor shorter rules, or to better align the features with our proposed rule types. Jermyn et al. (2025) report preliminary work in this direction, which could be adapted to our setting.

**Towards understanding the full model.** We envision this work as a step towards understanding the full transformer in terms of rules. Assuming we can accurately describe attention features, one next step is to characterize how these rules are used to form larger units of computation. These include: how single-head features combine to form multi-head features (i.e. "attention superposition"; Jermyn et al., 2023); how features are processed by feed-forward layers—a subject addressed by Dunefsky et al. (2024a); and how features are composed accross layers (e.g. as in induction heads; Olsson et al., 2022). This granular, rule-based characterization could be complemented with more abstract descriptions of higher-level computations, such as natural language.

## 6 RELATED WORK

**Explaining transformer features with SAEs.** The first step in the mechanisitic interpretability pipeline is to decompose dense transformer hidden representation into a dictionary of interpretable features (e.g. Ameisen et al., 2025). Recent work has made progress towards this goal by using sparse dictionary-learning methods, in particular sparse auto-encoders (SAEs; e.g. Bricken et al., 2023). These activating examples can be visualized in a dashboard (Bricken et al., 2023) or summarized

automatically by a language model (Bills et al., 2023). However, natural language explanations are subjective (Huang et al., 2023), and exemplar-based explanations can be illusory if the reference dataset is not sufficiently diverse (Bolukbasi et al., 2021). Our goal in this work is to explain features mechanistically, in terms of explicit transformations of input features.

**Transformer feature circuits.** Our investigation extends a recent line of work on understanding how SAE features interact to form computational graphs, or "circuits." The concept of a transformer circuit was introduced by Elhage et al. (2021), who studied attention-only transformers. Elhage et al. (2021) proposed decomposing an attention layer into a "key-query" (KQ) circuit, which determines the attention pattern, and an "output-value" (OV) circuit, which determines how tokens that receive non-zero attention affect the output features. A similar approach may unearth SAE "feature circuits," by identifying the computational graph composed of features that are active for a given prompt (Marks et al., 2024; Dunefsky et al., 2024a; Ge et al., 2024; Ameisen et al., 2025). Although the attention pattern depends on the input, these methods treat it as a fixed pattern, allowing a linear decomposition of a layer's activations in terms of those of the previous layers. Our goal is to extend the feature-circuit approach to the attention mechanism itself.

**Characterizing attention features.** Some work has attempted to characterize attention features in terms of human-understandable formalisms. Theoretically, some work has attempted to characterize the kinds of features transformers can express in terms of programs (Weiss et al., 2021), logical frameworks (e.g. Merrill & Sabharwal, 2023), or formal languages and algebraic automata (Liu et al., 2023; Yang et al., 2024); see Strobl et al. (2024) for a survey. These approaches have been used to develop intrinsically interpretable variants of transformers (Friedman et al., 2023), but there has been relatively little work to use these formalisms to extract rules from black-box transformers. Elhage et al. (2021) identify some attention motifs, such as the "skip-gram" pattern `...[A]...[B]`, which is activated when token `B` is preceded by token `A`. We will extend this approach to describe patterns of SAE features, and consider other, more sophisticated features. Most similar to our work, Ge et al. (2024) decompose attention scores into interactions between SAE features and provide case studies illustrating how this approach can be used for local and global analysis of attention behaviors. In this work, we aim to provide a more general methodology for formalizing and extracting attention rules, and to evaluate how well these rules can approximate features in an empirical language model.

**Rule-based descriptions of neural networks.** There has long been interest in extracting formal, rule-based descriptions of neural networks (Andrews et al., 1995; Jacobsson, 2005; Mekkaoui et al., 2023). Although rule extraction has seen some success with simpler architectures like RNNs, extending this approach to transformers presents unique challenges. Unlike RNNs—which can be understood in terms of finite-state automata or regular expressions (Wang et al., 2018; Weiss et al., 2018)—the attention layer in transformers enables a richer class of computational patterns that cannot be captured by traditional formalisms. Works on "transformer programming languages " (Weiss et al., 2021; Yang et al., 2024), modal logic (Merrill & Sabharwal, 2023), and circuit complexity (Hahn, 2020; Hao et al., 2022) provide insight into what transformers could learn in theory by enumerating some classes of languages that transformers provably can or cannot represent. However, they focus on expressivity bounds rather than serving as interpretability targets—that is, they do not provide methods to extract rules that describe what a given black-box transformer has learned in practice.

# 7 CONCLUSION

In this work, we have taken initial steps towards automatically characterizing attention layers with rules. We outlined some initial formalisms for attention rules, developed methods for rule extraction, and measured how well these rules can approximate features in an empirical language model. While relatively simple rules can achieve reasonable approximation quality of some features, we also identified cases that are more challenging to describe with concise rules. Future work is needed to extend this approach to cover more types of rules, extract better underlying features, and build on these features to understand larger units of model computation.

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

## A APPENDIX

### A.1 SAE TRAINING DETAILS

As our testbed for extracting attention rules, we train SAEs on the outputs of every attention head in GPT-2 Small (Radford et al., 2019). Our training setup follows Kissane et al. (2024), with the difference that we train a separate SAE for each attention head (rather than training an SAE on the concatenated outputs of all attention heads in each layer). We train SAEs with a dictionary size of 2,048 on 2 billion tokens from OpenWebText (Gokaslan et al., 2019). We train on sequences of 64 tokens each, sampled so that each sequence consists of 64 contiguous tokens from a single document. SAEs are trained with the Adam optimizer (Kingma & Ba, 2014) with learning rate 0.0012, $\beta_1 = 0.9$, and $\beta_2 = 0.99$, with a batch size of 4,096. Following Kissane et al. (2024), we use use the neuron re-sampling scheme described by Bricken et al. (2023) to re-initialize neurons that do not fire for some number of iterations: every 25,000, 50,000, 75,000 and 100,000, neurons that have not fired for the last 12,500 steps are re-initialized. Each SAE is trained on

Our input features are obtained from an open-source SAE trained on the residual stream of GPT-2 Small (Bloom, 2024), which have a dictionary size of 24,576.

### A.2 DATA AND EVALUATION

**Collecting feature exemplars.** To evaluate rule extraction methods, we collect datasets of input sequences that activate our attention output features. We follow prior work (e.g Bills et al., 2023; Choi et al., 2024) and identify exemplars by calculating feature activations for 50,000 sequences drawn from the same data used to train the SAES. Following Bills et al. (2023), we randomly sample sequences of 64 tokens, without crossing document boundaries (i.e. each sequence consists of a contiguous subsequence of a document). We randomly sample 100 features for each attention head, considering only features that are (1) active in at least 150 input sequences, and (2) inactive in at least 150 input sequences. We create a dataset for each feature by selecting the 150 sequences that contain the highest activations, and randomly sampling 150 sequences for which the feature is inactive, and randomly partitioning the positive and negative examples into equal-sized train, validation, and test sets. We evaluate our rule extraction methods at predicting the activation for a single position $t$ in each sequence. For positive examples, $t$ is the position with the maximum activation. For negative examples, $t$ is randomly sampled from $(1, 64]$.

**Evaluation.** To measure how well rules approximate feature activations, we use a binary evaluation metric, roughly following Foote et al. (2023). Specifically, our feature datasets consist of an equal number of positive examples (with activations greater than 0), and negative examples (with activations equal to 0). Our rule extraction method outputs a scalar value corresponding to the predicted activation value. We threshold both the actual activation values and the predicted values to correspond to a positive prediction if the value is greater than 0 and a negative prediction otherwise, and measure the precision, recall, and F1. In the event that the rule extraction method does not predict any positive examples, we define the precision as 1.

We adopt this binary evaluation metric for simplicity, and because it is easy to interpret. But binarizing the feature activations discards potentially meaningful information. Other work on evaluating natural language explanations of neurons has reported the Spearman correlation between true and predicted activations (e.g. Bills et al., 2023; Bricken et al., 2023). A second limitation of our evaluation is that we consider only the examples with the highest activations for a given feature, along with randomly sampled negative examples. As noted by some prior work (Huang et al., 2023; Gao et al., 2024), this evaluation favors recall over precision, because we are unlikely to sample inputs that resemble positive examples but do not activate the feature. One possible approach to include inputs with a range of activation values. For example, Bricken et al. (2023) divide the activations into quantiles and evaluate feature simulations within each quantile.

### A.3 RULE EXTRACTION

In this section, we provide additional details about our methods for identifying a small number of input features to use to explain a given output feature, introduced in Section 3.2.

| DFA | Feature activations |
|---|---|
| atsun and witnesses reported seeing a matching vehicle veering off the highway and stopping | atsun and witnesses reported seeing a matching vehicle veering off the highway and stopping |
| to our source, the new sports sedan will target 40-something buyers who hon | to our source, the new sports sedan will target 40-something buyers who hon |
| by Tesla. As part of a community of Tesla owners, whenever Aut op ilot | by Tesla. As part of a community of Tesla owners, whenever Autop ilot |

(a) Top activating sequences for L0H0.1476.

| Key | Val. score | Query | Attn. score |
|---|---|---|---|
| _vehicle | 0.145 | _sober | 0.141 |
| | | _cruise | 0.128 |
| | | _accelerate | 0.126 |
| _vehicles | 0.139 | _sober | 0.122 |
| | | _cruise | 0.11 |
| | | _instrument | 0.109 |
| _truck | 0.096 | _sober | 0.118 |
| | | _seat | 0.097 |
| | | _Bra | 0.097 |

(b) Top scoring key/query pairs.

Figure 8: Sequences that activate a layer-0 attention feature in GPT-2 small (8a) and the highest-scoring pairs of key and query input features associated with this feature (8b). This feature activates in more diverse contexts, and there are more key/query pairs with relatively high scores.

Let $\mathbf{y}_t \in \mathbb{R}^{d_{\text{head}}}$ denote the output of an attention head, and $g(\mathbf{y}_t) = \sigma(\mathbf{y}_t^\top \mathbf{u})$ denote a single output feature from an attention-output SAE, where $\mathbf{u} \in \mathbb{R}^{d_{\text{head}}}$ is the encoder vector associated with feature $g$. For a feature $i$ in the input SAE, we define value-score$(i) = \mathbf{d}_i^\top \mathbf{W}_V^\top \mathbf{u}$. For any query feature $i$ and key feature $j$, we define attention-score$(i,j) = \mathbf{d}_i^\top \mathbf{W}_Q^\top \mathbf{W}_K \mathbf{d}_j$.

Our input SAEs have a dictionary size of $24,576$, meaning there are potentially $24,576^2$ pairs of key and query features relevant to characterizing a given output feature. We consider two methods for identifying a small number of input features. For both methods, we first use a weight-based procedure for identifying a relatively small number of candidate features. First, we pick the 100 key features $k$ with the highest values of value-score$(i)$. Second, for each of these key features $k$, we pick the 100 query features maximizing attention-score$(q,k)$, resulting in 10,000 key/query pairs. Then we use either a weight-based or gradient-based method for selecting a small number of key/query pairs from the list of candidates. For our weight-based method, we simply the list of pairs in decreasing order of attention-score$(q,k) *$ value-score$(k)$.

For our gradient-based method, we introduce a mask $m_{i,j} = 1$ for each pair of query and key features, and we define a masked attention score attention-score$(i,j) = m_{i,j} \mathbf{d}_i^\top \mathbf{W}_Q^\top \mathbf{W}_K \mathbf{d}_j$. We calculate the feature activations for all of the prefixes in our training set using the masked attention scores, and calculate the average gradient of the output feature for each mask: $dg/dm_{i,j}$. We pick the the key/query pairs with the highest importance scores.

## A.4 Additional Results

**Skip-gram Examples.** Fig. 8 shows examples of a feature from a layer-0 attention feature and the corresponding skip-grams of key and query pairs. Fig. 8a indicates that this feature fires in a variety of contexts related to automobiles. Compared to the example, illustrated in Fig. 5, this rule is relatively longer—because it consists of more terms—but it remains relatively simple to understand because the terms do not interact.

**Absence Rule Examples.** Fig. 9 shows the percentage of features at each layer for which we can find a *distractor key*. We say that a feature has a distractor key if there is a feature $k'$ such that value-score$(k) < 0$ and attention-score$(q,k') >$ attention-score$(q,k)$, where $k$ and $q$ are the features maximizing attention-score$(q,k) *$ value-score$(k)$. Fig. 10 shows an example of a feature with such a distractor key, where $q$ fires on the token "com", $k$ fires on the token "://", and $k'$ fires on the token "twitter". The feature normally fires at the end of URLs. If "twitter" appears in the sequence, it attracts attention away from the "://" token and suppresses the output feature.

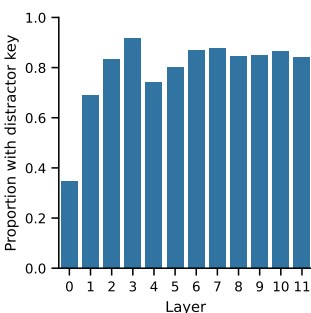

Figure 9: The percentage of attention output feature that have a distractor key. Many attention output features are associated with absence rules, with absence rules growing more common at higher layers.

| Attention | Feature activations |
|---|---|
| to the interview on CNBC . com : http :// video . cn bc . com | to the interview on CNBC.com:     http:// video . cn bc. com |
| <lendoftextl>   \n   Ret weet : http :// twitter . com | <lendoftextl>\nRetweet: http://twitter.com |

(a) Activating and non-activating sequences for L0H0.429.

| Key | Val. score | Query | Attn. score |
|---|---|---|---|
| :// | 0.119 | com | 0.079 |
| twitter | -0.007 | com | 0.097 |

(b) Top scoring key/query pair and a *distractor* key.

Figure 10: An example of a feature exhibiting an absence rule. The output feature is activated when the input sequence matches the pattern . . . [://] . . . [com], unless the sequence also contains the distractor feature "twitter".