# OpenReview forum: "Extracting Rule-based Descriptions of Attention Features in Transformers"
_ICLR.cc/2026/Conference — ICLR 2026 Conference Withdrawn Submission_

### Official Review · Reviewer_ytiW · 2025-10-28

**Soundness:** 2
**Presentation:** 2
**Contribution:** 2
**Rating:** 4
**Confidence:** 2

**Summary:**

This paper provides a mechanistic interpretability technique for transformers that takes into account the attention mechanism by analyzing the ways in which feature vectors of a sparse autoencoder (SAE) interact within the attention mechanism to promote or discourage certain outputs. This allows a practitioner to identify three kinds of rule embedded in a transformer: (1) skip-gram rules that promote an output based on the presence of a skip bigram, (2) absence rules, which are like (1) but discourage an output, and (3) counting rules, which are like a competition between rules of the form (1) and (2). The authors provide two ways of ranking query-key interactions in order to prune low-ranking ones: a weight-based one and a gradient-based one. They test their method on GPT-2 small. They evaluate their automatically extracted skip-gram rules on the binary task of predicting whether a feature is active for a given prefix, achieving F1 scores of about 70-80%. They also identify skip-grams that have distractor features which make them act like absence rules.

**Strengths:**

The paper is generally well-written and does a good job of contextualizing itself with respect to prior work. Section 3.1 does a good job of presenting the main idea of the paper, which appears to be an original and significant extension of mechanistic interpretability to the attention mechanism. The authors test multiple ranking strategies for pruning.

**Weaknesses:**

1. The method is limited in its expressivity; the types of rule extracted are quite simple, and it does not appear to be straightforward to extend it to take into account other parts of the transformer architecture besides the attention mechanism, such as the feedforward layers. However, this may not be a very serious issue, as mechanistic interpretability methods typically rely on these simplifications, and 3.1 does a good job of justifying the types of rules they study.
1. I'm not sure how to interpret the significance of the results in Fig 3, because there is no baseline. Is 80% F1 good or bad?
1. There don't seem to be any results on counting rules, despite their prominence in the exposition of the paper. This aspect of the paper is incomplete.
1. GPT-2 is pretty old; it would be nice to see this method evaluated on more than one LLM, including a newer one.
1. Although I found some parts of the paper, like 3.1, to be quite clear and helpful, other parts are hard to follow. In particular, I'm a bit lost as to exactly how the experiments and evaluation in Section 4 work. For the binary classification task, what are you using as the ground-truth labels?

**Questions:**

1. Eq (2): Why are the last two steps identical?
1. Fig 5: There seems to be an off-by-one lag after "If" in the feature activations, and there's something strange going on with "Both." What's going on there?

---

### Official Review · Reviewer_DttW · 2025-10-31

**Soundness:** 2
**Presentation:** 2
**Contribution:** 2
**Rating:** 4
**Confidence:** 3

**Summary:**

This paper proposes a method to interpret attention-layer Sparse Autoencoder (SAE) features by extracting symbolic, human-readable rules. The paper focus on skip-grams, absence rules, and counting rules. This approach differs from current methods to interpret SAEs, which are mainly focused on observing the activation of a SAE feature in tokens of the training corpus and using autointerp methods to give meaning to them. The paper introduces $\mathcal{S}(j)$ to quantify interactions between input-output features (OV circuit) and $\mathcal{A}(j, k)$ to quantify the interaction between a pair $(j, k)$ of input features (QK circuit). Applying this method to GPT-2 small, the authors show that simple skip-gram rules can approximate many features with reasonable fidelity and that absence rules are common

**Strengths:**

- The paper proposes a new method to interpret SAEs features. It differs from previous work especially in the treatment of the QK features, trying to analyze feature interactions that contributes to the attention pattern formation.
- The method is capable of finding interesting set of features that it may be hard to do with autointerp.

**Weaknesses:**

1. Some of the claims relies on weak evidence:
    - The evidence for counting rules is a single qualitative example (Figure 7).
    - In the absence rules analysis, the counterfactual validation (Fig 6c) is limited to only the first attention layer.
2. The entire evaluation is limited to one model, GPT-2 small. It is not clear if this methodology will apply to larger models.
3. The method for ranking rules relies on heuristics, like picking the "top 100" features to reduce the search space. These choices are not well-justified, and it is not guaranteed that they are generalizable to other models.
4. The F1-score evaluation is hard to interpret. It's run on a simplified dataset (top 100 activations vs. 100 zero-activations). The paper itself notes in the appendix that this setup favors recall, so it's hard to tell if an F1-score of ~0.75 is actually good.

**Questions:**

1. Can you provide quantitative evidence (e.g., F1 scores, prevalence) for counting rules?
2. Why was the validation for absence rules limited to Layer 1? Is this experiment intractable on higher layers?
3. For the gradient-based ranking, did you consider using more robust attribution methods like Integrated Gradients (IG) instead of simple gradients?
4. I found the notation is a bit confusing.
    - In Section 3.1, the sum with $i \leq t$ and $j=1$ to $n$ is confusing. Using a double sum would make it easier to read (one sum is related to the tokens, while another one is related to the $n$ SAE features), or using $1 \leq j \leq n$ instead of $j=1$, removing the $n$ from the top of the summation.
    - In Section 3.2, in the “Ranking query-key rules” paragraph, it is said that you first pick the 100 key features $j$ with highest value of $\mathcal{S}(j)$, then for each of these key features $j$, you pick the 100 query features maximizing $\mathcal{A}(j, k)$. In my understanding, for $\mathcal{A}(j, k)$, $j$ is a query feature, while $k$ is a key feature. Shouldn’t be the case that you pick the 100 query features maxizing  $\mathcal{A}(k, j)$ instead?

---

### Official Review · Reviewer_7pMq · 2025-10-31

**Soundness:** 2
**Presentation:** 1
**Contribution:** 2
**Rating:** 2
**Confidence:** 3

**Summary:**

The paper describes an approach to find symbolic templates (aka rules) which activate specific patterns in the attention heads of transformer based language model. The idea builds on the sparse-autoencoder features, and uses them to identify three types of rules: n-grams, absence rules and counting rules in the GPT-2 model. The rules found are illustrated with examples, as well as evaluated quantitatively.

**Strengths:**

- The paper falls within the mechanistic interpretability tradition and builds on the work of Kissane et al 2024 with sparse autoencoders. The suggested method for finding rules goes beyond manual exemplar inspection and is thus able to find rules which are hard to identify manually
- While the details of the methodology are somewhat obscure in places the general approach seems reasonable and sound.
- The results may be of some interest given the amount of attention to mechanistic interpretability.

**Weaknesses:**

- The main contribution is the procedure for finding rules: while it's an improvement over manual examplar examination, it is still based on quite strong priors and hard-coded search patterns and assumptions.
- The advertised symbolic and interpretable nature of the found rules only really applies to the bottom transformer layer where inputs can be directly linked to input tokens. In layers above, the interpretation of the features and rules becomes increasingly murky.
- The presentation relies heavily on detailed familiarity with previous work and is thus very hard to follow as a self-contained paper.
- The focus is exclusively on single attention heads, while transformers have many other components which impact the computations. Thus the scope is very narrow.

**Questions:**

- How are feature activations and direct feature attributions related, and why are you presenting both in your results?
- How should I think about interpreting these rules in layers > 1 of the transformer, where inputs are no longer tokens?

---

### Official Review · Reviewer_hiA6 · 2025-11-01

**Soundness:** 3
**Presentation:** 3
**Contribution:** 2
**Rating:** 6
**Confidence:** 3

**Summary:**

The paper studies attention-head interpretability at the level of SAE features. To this end. the authors train sparse autoencoders on attention head outputs of GPT-2 small (using an existing setup on open web text (OWT) and then aim to identify attention-output features by extracting rule-like patterns from attention’s query–key (QK) and value (OV). To this end they 1) collect a feature-specific dataset from 50k OWT sequences (positives = top activations of g, negatives = non-activations) for each attention-output feature they identified, and 2) rank candidate (query-feature, key-feature) pairs either by a weight score (\text{QK}\times \text{OV}) or by a gradient mask score. From the top pairs they induce three rule types:
study three types:
	(1) skip-gram rules of the form “[Canadian city] . . . speaks → English”,
	(2) absence rules of the form “[Montreal] . . . speaks ̸→ English,” and
	(3) counting rules that toggle only when the count of a word exceeds a certain value or the count of another word.
They use GPT-2 small and show that many early-layer features can be approximated by less than 100 rules, that distractor rules are common, and that count-sensitive rules already show up in low layers.

**Strengths:**

- The paper looks at feature level for behaviors, such as skip-gram patterns, that are frequently studied in circuit analysis, providing an interesting view from the SAE lens, making the analysis better scalable.
- It identifies the discovered rules on a broad set of sequences, indicating robustness of these features.
- It focuses on a clearly defined set of rules that can be sensibly studied.

**Weaknesses:**

- The evaluation, while on a broad set of samples is qualitatively limited to binary activations;
- The paper does not compare to traditional circuit analysis approaches, which could analyse the same patterns, showing how much they overlap.
- Similarly, there are no ablations on other methods beyond SAEs
- While the paper evaluates how often the patterns are present in the collected data, it would be interesting to see how they are present in real-world tasks.
- The approach uses a single, small model and the open web text corpus, so generalizability on to larger/different models and also corpora is limited here.

**Questions:**

How would the rule extracting method perform on circuit heads (such as induction, name-mover, IOI-style)?
How sensitive are your results to model size and data selection to train the SAE?

---

### Note · Authors · 2025-11-26

I have read and agree with the venue's withdrawal policy on behalf of myself and my co-authors.